# Effect of Cardiac Rehabilitation on Left Ventricular Diastolic Function in Patients with Acute Myocardial Infarction

**DOI:** 10.3390/jcm10102088

**Published:** 2021-05-13

**Authors:** Jae-Hwan Lee, Jungai Kim, Byung Joo Sun, Sung Ju Jee, Jae-Hyeong Park

**Affiliations:** 1Division of Cardiology in Internal Medicine, Chungnam National University Sejong Hospital, Chungnam National University School of Medicine, Sejong 30099, Korea; myheart@cnuh.co.kr (J.-H.L.); jakim671209@cnuh.co.kr (J.K.); 2Department of Cardiology in Internal Medicine, Chungnam National University Hospital, Chungnam National University School of Medicine, Daejeon 35015, Korea; bjcardio@hanmail.net; 3Department of Rehabilitation Medicine, Chungnam National University Hospital, Chungnam National University School of Medicine, Daejeon 35015, Korea; drjeesungju@cnuh.co.kr

**Keywords:** acute myocardial infarction, prognosis, cardiac rehabilitation, diastolic function

## Abstract

Cardiac rehabilitation (CR) improves symptoms and survival in patients with acute myocardial infarction (AMI). We studied the change of diastolic function and its prognostic impact after CR. After reviewing all consecutive AMI patients from January 2012 to October 2015, we analyzed 405 patients (mean, 63.7 ± 11.7 years; 300 males) with baseline and follow-up echocardiographic examinations. We divided them into three groups according to their CR sessions: No-CR group (*n* = 225), insufficient-CR group (CR < 6 sessions, *n* = 117) and CR group (CR ≥ 6 sessions, *n* = 63). We compared echocardiographic parameters of diastolic dysfunction including E/e’ ratio > 14, septal e’ velocity < 7 cm/s, left atrial volume index (LAVI) > 34 mL/m^2^, and maximal TR velocity > 2.8 m/s. At baseline, there were no significant differences in all echocardiographic parameters among the three groups. At follow-up echocardiographic examination, mitral annular e’ and a’ velocities were higher in the CR group (*p* = 0.024, and *p* = 0.009, respectively), and mitral E/e’ ratio was significantly lower (*p* = 0.009) in the CR group. The total number of echocardiographic parameters of diastolic dysfunction at the baseline echocardiography was similar (1.29 vs. 1.41 vs. 1.52, *p* = 0.358). However, the CR group showed the lowest number of diastolic parameters at the follow-up echocardiography (1.05 vs. 1.32 vs. 1.50, *p* = 0.017). There was a significant difference between the No-CR group and CR group (*p* = 0.021). The presence of CR was a significant determinant of major adverse cardiovascular events in the univariate analysis (HR = 0.606, *p* = 0.049). However, the significance disappeared in the multivariate analysis (HR = 0.738, *p* = 0.249). In conclusion, the CR was significantly associated with favorable diastolic function, with the highest mitral e’ and a’ velocity, and the lowest mitral E/e’ ratio and total number of echocardiographic parameters of diastolic dysfunction at the follow-up echocardiographic examinations in AMI patients.

## 1. Introduction

Acute myocardial infarction (AMI) is the most common cause of acute heart failure, and prompt management of the acute phase in AMI patients is mandatory to prevent the transition of acute heart failure to chronic heart failure [1]. Along with coronary revascularization and anti-ischemic pharmacotherapy, cardiac rehabilitation (CR) is the best adjunctive modality associated with the improvement of symptoms and survival in patients with AMI [2,3]. Comprehensive CR is an out-patient disease management program that reduces CV mortality by approximately 25% and hospital readmissions by 18% [4,5]. Due to its proven clinical benefits and cost-effectiveness, recent treatment guidelines recommend CR after AMI as the class I recommendation. CR includes coordinated activities necessary for favorably influencing the underlying causes of cardiovascular diseases; for facilitating optimal physical, mental, and social conditions; and for enabling patients to preserve or find the best possible way to function within their community [6]. Physical activity is often the most important part of CR programs. The CR-associated exercise program can improve cardiac function in patients with coronary artery diseases [7,8,9]. CR can also improve clinical outcomes in patients with heart failure (HF), including end-stage HF treated with a left ventricular assist device system [10].

LV diastolic dysfunction can be associated with exercise intolerance, and it was associated with frailty and poor prognosis in elderly patients with acute coronary syndromes [11]. However, the effect of exercise training on the left ventricular (LV) diastolic function in patients with AMI remains controversial [7,9,12]. Thus, we studied the influence of CR on diastolic dysfunction and its prognostic influence in AMI patients.

## 2. Materials and Methods

### 2.1. Study Population

We retrospectively reviewed all consecutive type 1 MI patients from January 2012 to October 2015. Baseline clinical data were obtained from their medical records. We defined type 1 MI-based elevation and/or fall of cardiac troponin values and with at least one of their clinical presentations including symptoms of acute myocardial ischemia, electrocardiographic findings including new ischemic changes or new pathological Q waves, imaging evidence of a new loss of viable myocardium or appearance of regional wall motion abnormality consistent with coronary territories, and identification of the thrombus by coronary angiography [13]. We checked for incidences of death in the medical records of patients who had been regularly followed-up. In patients without regular follow-ups, we identified major adverse cardiac events (MACE) including deaths, recurrence of MI, angina and revascularization, admissions for heart failure, and stroke or transient ischemic attack by speaking with the patients or their relatives over telephone, or data from the Korean national insurance service. 

This study was approved by our institutional review board (IRB, no. 2016-04-034). We performed this study according to the ethical standards laid down in the 1964 Declaration of Helsinki and its later amendments. This was a retrospective study and many of the subjects had already died; therefore, the IRB waived gathering informed consent from the participants.

### 2.2. Echocardiographic Analysis 

Echocardiographic data were acquired from digitally stored echocardiographic images of the patients. LV dimensions were calculated using the parasternal long-axis view. LV systolic function was estimated from LV ejection fraction (LVEF). We used a modified biplane Simpson’s method to calculate LVEF with an apical four chamber and an apical two chamber view. LV diastolic function was assessed by considering key echocardiographic variables including mitral E and A velocities, mitral annular e’ velocity, mitral annular a’ velocity, mitral E/e’ ratio, peak velocity of the tricuspid regurgitation jet (TR Vmax), and left atrial maximum volume (LAV), as recommended by the American Society of Echocardiography [14]. LA volume was calculated by the modified Simpson’s method with apical four-chamber and apical two-chamber views. LAV was indexed based on the body surface area and expressed as the LAV index (LAVI). 

We checked the presence of four recommended variables to identify elevated LV filling pressure. The abnormal cutoff values of these four variables are septal annular e’ velocity < 7 cm/s, septal E/e’ ratio > 14, LAVI > 34 mL/m^2^, and peak velocity of tricuspid regurgitation (TR Vmax) > 2.8 m/s. We assessed the degree of diastolic dysfunction following the algorithm proposed in the latest guideline from the American Society of Echocardiography and the European Association of Cardiovascular Imaging [14]. We classified our patients into four groups: normal, grade 1, grade 2, and grade 3 diastolic dysfunctions. Patients with normal or grade 1 diastolic dysfunction were considered to have normal LA pressure, and those with grade 2 and grade 3 diastolic dysfunction with having elevated LA pressure.

### 2.3. Cardiac Rehabilitation Program

We performed a 6-week CR program with graded exercise tests at the cardiac rehabilitation clinic during the first visit of these patients after their discharge. The exercise duration was 50 min, including 10 min of warm-up, 30 min of main exercise (15 min of treadmill exercise and 15 min of ergometer), and 10 min of cool down. The exercise session was conducted 3 times per week for a total of 18 sessions. For the graded exercise test, symptom-limited exercise was performed according to a modified Bruce protocol. The target heart rate was calculated using the Karvonen formula; the target heart rate was calculated at 60% of the maximal heart rate during the first 2 weeks, at 70% during the next 2 weeks, and at 85% during the last 2 weeks. 

An electrocardiogram was monitored to determine the heart rate and a probable abnormal change in it during the exercise. Borg’s scale was used to evaluate the symptoms of patients along with the rate of perceived exertion. 

We divided our study population into three groups according to CR intervention: No-CR group, insufficient-CR group (CR < 6 sessions), and CR group (CR ≥ 6 sessions). 

### 2.4. Statistical Analysis

We expressed categorical variables as frequencies and percentages and continuous variables as the mean ± standard deviation. We compared categorical variables using the chi-squared test. For continuous variables, we checked the distribution of the variables by the Shapiro–Wilk test. If the variable showed normal distribution, we used an analysis of variance (ANOVA) test to find statistical differences among the three groups and a *t*-test between the two groups. We used the Kruskal–Wallis test among three groups and the Mann–Whitney test between two groups of the variables without normal distribution. Moreover, the baseline and the follow-up echocardiographic parameters were compared using the Wilcoxon signed ranks test. 

A Cox proportional hazards model was used to find factors associated with the occurrence of adverse clinical events. Multivariate analysis was performed using statistically significant variables found in the univariate analysis. We performed the multivariate analysis with bootstrapping with a sample size of 1000 to avoid multicollinearity. A two-tailed *p*-value of <0.05 was considered statistically significant. All statistical analyses were performed using SPSS version 22.0 (IBM, Chicago, IL, USA) and MedCalc version 12.3.0.0 (MedCalc Software, Mariakerke, Belgium).

## 3. Results

### 3.1. Characteristics of Study Population

After reviewing all consecutive type 1 MI patients from January 2012 to October 2015, we included 405 patients with baseline and follow-up echocardiographic examinations. The mean interval between the admission date to echocardiographic examination was 1.6 ± 1.8 days (interval: 0–14 days).

We divided our study subjects into three groups depending on whether they received CR; No-CR group (*n* = 225), insufficient-CR group (*n* = 117), and CR group (*n* = 63). Their baseline clinical and echocardiographic parameters are expressed in Table 1. Age was significantly higher in the No-CR group (65.2 ± 12.4 years vs. 62.9 ± 11.2 years vs. 61.4 ± 9.5, *p* = 0.017). For cardiovascular risk factors, only hypertension (52.4% vs. 44.4% vs. 33.3%, *p* = 0.021) was significantly higher in the No-CR group. The percentage of non-ST-segment elevation myocardial infarction was similar in the two groups (42.2% vs. 40.2% vs. 30.2%, *p* = 0.223). However, patients with Killip class III/IV were more frequent in the No-CR group (7.6% vs. 0% vs. 4.8%, *p* = 0.009). 

Regarding echocardiographic parameters, there was no statistical difference in LVEF (47.0 ± 11.5% vs. 47.9 ± 9.3% vs. 47.6 ± 9.4%, *p* = 0.986) or regional wall motion abnormality assessed by the wall motion score index (1.55 ± 0.42 vs. 1.51 ± 0.35 vs. 1.52 ± 0.32, *p* = 0.850). Additionally, there were no statistically significant differences of diastolic parameters among three groups, except LAVI (*p* = 0.043) and E/e’ ratio (*p* = 0.045). Coronary angiography and percutaneous coronary intervention were performed in 400 patients (98.6%). The left anterior descending (LAD) coronary artery was the most common culprit lesion. Complete revascularization was achieved in 385 patients (96.3%), and there was no statistical difference in the success rate among the three groups.

### 3.2. Follow-Up Echocardiography

Follow-up echocardiographic examinations were performed for a mean duration of 18.0 ± 15.1 months, and the intervals between the last CR and the follow-up echocardiography was 17.1 ± 14.9 months in the insufficient-CR group and 13.4 ± 11.5 months in the CR group. Findings of the follow-up echocardiography and a comparison between the baseline and follow-up echocardiography findings are summarized in Table 2. At follow-up echocardiography, LVEF was significantly improved in all three groups (No-CR group: 47.0 ± 11.5% to 51.0 ± 12.1%, *p* < 0.001, insufficient-CR group: 47.9 ± 9.3% to 52.0 ± 10.5%, *p* < 0.001, CR group: 47.6 ± 9.4% to 53.9 ± 11.0%, *p* < 0.001). LV end-diastolic dimension was significantly increased in the No-CR group (47.7 ± 7.0 to 48.5 ± 7.1%, *p* = 0.042). LAVI was significantly decreased in all the three groups (No-CR group: 37.0 ± 17.1 mL/m^2^ to 35.1 ± 19.2 mL/m^2^, *p* = 0.001, insufficient-CR group: 34.5 ± 16.6 mL/m^2^ to 31.6 ± 14.2 mL/m^2^, *p* = 0.027, and CR group: 31.4 ± 9.7 mL/m^2^ to 29.4 ± 10.6 mL/m^2^, *p* = 0.049). In addition, mitral E velocity was decreased in the No-CR group (68.5 ± 22.6 cm/s to 63.4 ± 20.1 cm/s, *p* = 0.023) and the insufficient-CR group (70.3 ± 23.9 cm/s to 64.7 ± 26.0 cm/s, *p* = 0.006). Mitral A velocity was decreased in the No-CR group (80.1 ± 21.5 cm/s to 75.9 ± 22.8 cm/s, *p* = 0.005) and the insufficient-CR group (76.1 ± 20.8 cm/s to 73.2 ± 20.5 cm/s, *p* = 0.019). However, mitral E and A velocity did not change in the CR group (*p* = 0.192 and *p* = 0.795, respectively). 

In the comparison of the three groups, mitral annular e’ and a’ velocities were higher in the CR group (*p* = 0.024, and *p* = 0.009, respectively), and the mitral E/e’ ratio was significantly lower (*p* = 0.009) in the CR group.

In the comparison of the three groups, LAVI was significantly lower (*p* = 0.026) in the CR group.

### 3.3. Echocardiographic Variables of Diastolic Dysfunction

Table 3 describes the presence of echocardiographic variables of diastolic dysfunction in the three groups. There was no statistical significance of diastolic parameters among the three groups at the baseline echocardiographic examinations, including the estimation of LV filling pressures. At the follow-up echocardiographic examinations, the number of patients with LAVI > 34 mL/m^2^ was significantly higher in the CR group (*p* = 0.042). Additionally, the total number of diastolic variables was significantly lower in the CR group (*p* = 0.017). The statistical differences of LAVI > 34 mL/m^2^ and total number of diastolic parameters mainly occurred between the No-CR and CR groups (*p* = 0.018 and *p* = 0.006, respectively). At the follow-up echocardiographic examinations, the presence of normal LV filling pressure was higher in the CR group. However, there was a marginal statistical significance (*p* = 0.083).

### 3.4. Adverse Clinical Outcomes during the Follow-Up Period

During the follow-up period (mean, 72.8 ± 24.4 months), there were 190 occurrences of MACE (comprising 23 deaths, 8 admissions for AMI recurrence, 62 patients with angina and revascularization, 36 admissions for heart failure, and 25 patients with stroke or transient ischemic attack) across 117 patients. 

The results of univariate and multivariate analyses for the prediction of MACE are detailed in Table 4. 

In the univariate analysis, statistically significant variables included age (HR = 1.029, *p* < 0.001), hypertension (HR = 1.590, *p* = 0.004), diabetes (HR = 1.675, *p* = 0.002), baseline LAVI (HR = 1.013, *p* = 0.004), baseline E/e’ ratio (HR = 1.047, *p* < 0.001), total number of diastolic parameters at the baseline echocardiography (HR = 1.210, *p* = 0.041), follow-up LV end-systolic volume (HR = 1.008, *p* = 0.011), follow-up LAVI (HR = 1.020, *p* < 0.001), follow-up E/e’ ratio (HR = 1.032, *p* < 0.001), and total number of diastolic parameters at the follow-up echocardiography (HR = 1.418, *p* < 0.001). Interestingly, the presence of CR was a significant determinant of MACE (HR = 0.790, *p* = 0.049). In the multivariate analysis, age (HR = 1.035, *p* < 0.001), diabetes (HR = 1.627, *p* = 0.005), and total number of diastolic parameters at the follow-up echocardiography (HR = 1.255, *p* = 0.004) were the significant determinants of MACE. However, the presence of CR was not a significant determinant of MACE after the multivariate analysis (HR = 0.738, *p* = 0.249).

## 4. Discussion

In this study, we compared the follow-up and the baseline echocardiographic variables in patients with AMI. We showed that the CR group statistically had the highest mitral e’ and a’ velocities, the lowest mitral E/e’ ratio, and the lowest total number of echocardiographic parameters of diastolic dysfunction at the follow-up echocardiographic examinations in patients with AMI.

CR can reduce cardiovascular mortality and morbidity in patients with AMI. It improves exercise capacity and exerts beneficial effects on cardiovascular risk factors, including lipid profiles and insulin sensitivity [15]. It can also improve LV systolic function in patients with AMI [16,17]. Giannuzzi et al. reported a significant improvement in LVEF after a six-month exercise training program (from 34 ± 5% to 38 ± 8%, *p* < 0.01), but no improvement was noted in the control group (from 34 ± 5% to 33 ± 7%, *p* = nonspecific) [16]. Kim et al. showed a significant increase in LVEF (from 55.5 ± 7.8% to 59.6 ± 9.2%, *p* = 0.02) after a six-week exercise program [17]. Our study demonstrated results similar to those of previous studies, and the CR group showed a significant improvement in LVEF (47.6 ± 9.4% to 53.9 ± 11.0%, *p* < 0.001). A significant improvement in LVEF was also noted in the No-CR group (47.0 ± 11.5% to 52.8 ± 22.3%, *p* < 0.001) and in the insufficient-CR group (47.9 ± 9.3% to 52.0 ± 10.5%, *p* < 0.001). This could be attributed to the successful percutaneous coronary intervention, the use of anti-ischemic medications, and treatment for favorable LV remodeling. 

CR can improve diastolic function in patients with coronary artery disease. Wuthiwaropas et al. reported an improved diastolic function in half of their participants after receiving three-month CR, assessed by mitral E/e’ ratio [18]. Sandri et al. reported that a four-week regular exercise program was associated with an improvement in the E/e’ ratio in patients with heart failure with reduced ejection fraction [19]. 

Our study showed that baseline LAVI and mitral E/e’ ratio were the lowest in the CR group. These differences may have resulted from the age difference of the study groups. Additionally, age is another determinant of the worsening of diastolic function, and increased prevalence of LV diastolic dysfunction was associated with increasing age [20]. The active participants in the CR program were younger; therefore, they could have a lower prevalence of hypertension, LAVI, and mitral E/e’ ratio. Although there were significant differences of these diastolic parameters, the number of echocardiographic diastolic parameters was similar among the study groups. Younger age of the CR group can affect the change of LAVI. Cheng S. et al. showed that younger patients had better improvements of diastolic parameters in response to similar reductions in systolic blood pressure [21]. 

In our study, LAVI significantly decreased in all three groups in the follow-up echocardiographic examinations. LAVI was another parameter of LV diastolic dysfunction [22]; increased LAVI is a significant prognostic factor for MACE and all-cause mortality [23,24]. Additionally, LAVI can be associated with poor exercise capacity in patients with diastolic dysfunction [25]. The prognostic significance of LAVI has been confirmed in patients with AMI [26,27], and increased baseline LAVI was associated with a poor prognosis in our study. Patients with AMI who had an increased LAVI of >32 mL/m^2^ showed a poor five-year mortality outcome [27]. We demonstrated that higher baseline and follow-up LAVI values were good prognostic markers in our study patients. There have been limited data showing the change in LAVI as a prognostic marker in patients with AMI. Sakaguchi et al. [28] demonstrated that an increase in LAVI (>2.5 mL/m^2^) at the time of discharge was a predictor of MACE in first AMI patients. 

LAVI was also reduced in the No-CR and insufficient CR groups in our study. The decrease in LAVI could be associated with improved LV systolic function. In patients with severe heart failure, cardiac resynchronization therapy can improve LV systolic function along with LV diastolic function [29]. Percutaneous coronary intervention and anti-ischemic treatment improved LV systolic function, even in the No-CR group in our study. Additionally, there was a decrease in the proportion of grade 2 and 3 diastolic dysfunction in the No-CR group. 

CR was a significant determinant of MACE in the univariate analysis. However, the statistical significance was lost in the multivariate analysis. This result may come from the younger age, lower incidence of hypertension, and smaller LAVI when we included the patients at the baseline.

### Limitations 

This study has several limitations. Firstly, this was a retrospective study owing to a review of medical records. Although we recommended our subjects to regularly participated in the CR program, there was a substantial number of subjects who did not participate regularly owing to various reasons. Approximately 50% of the eligible patients for receiving CR were referred to CR in clinical practice, even in European countries [30]. In our study, there were more younger patients and a lower incidence of hypertension in the CR group at the baseline. Thus, they may have higher exercise capacity and motivation than those from other groups. It was possible that patients who were more enthusiastic about the CR program may have received additional treatments, including lifestyle advice and engagement with medical services. Secondly, we used the sum of diastolic parameters in our study. This approach has not been used in previous studies. The guidelines’ algorithm suggests that meeting more criteria suggests more advanced diastolic dysfunction. It should be validated in other studies. Thirdly, we did not exclude other factors, including significant valvular heart disease, which can affect LAVI. Additionally, we used only medical septal e’ velocity in the assessment of diastolic dysfunction. 

To solve these problems, well-controlled prospective studies will be needed to reveal the effect of CR on the LV diastolic function in AMI patients.

## 5. Conclusions

The CR group had the highest number of mitral annular e’ and a’ velocities, the lowest mitral E/e’ ratio, and total number of echocardiographic parameters of diastolic dysfunction at the follow-up echocardiographic examinations in AMI patients. However, the presence of CR was an insignificant predictor of long-term clinical outcomes.

## Figures and Tables

**Table 1 jcm-10-02088-t001:** Comparison of clinical and echocardiographic parameters according to the presence of cardiac rehabilitation (CR) therapy in discharged patients.

Variable	Total (*n* = 405)	No-CR Group (*n* = 225)	Insufficient-CR Group (*n* = 117)	CR Group (*n* = 63)	*p* Value
Age (year) ^+^	63.7 ± 11.7	65.2 ± 12.4	62.9 ± 11.2	61.4 ± 9.5	0.006
Male sex (%)	300 (74.1%)	164 (72.9%)	90 (76.9%)	46 (73.0%)	0.706
BMI (kg/m^2^)	24.0 ± 3.0	23.9 ± 3.3	24.1 ± 2.7	23.9 ± 3.0	0.747
Cardiovascular risk factors					
HTN (%) ^+^	191 (47.2%)	118 (52.4%)	52 (44.4%)	21 (33.3%)	0.021
DM (%)	127 (31.4%)	79 (35.1%)	33 (28.2%)	15 (23.8%)	0.159
Dyslipidemia (%)	18 (4.4%)	9 (4.0%)	7 (6.0%)	2 (3.2%)	0.722
Smoking (%)	159 (39.5%)	86 (38.6%)	49 (41.9%)	24 (38.1%)	0.415
Prior MI (%)	29 (7.2%)	20 (8.9%)	1 (1.3%)	8 (7.9%)	0.251
Ischemic heart disease (%)	37 (9.1%)	23 (10.2%)	10 (8.5%)	4 (6.3%)	0.619
Family history (%)	17 (4.1%)	8 (3.5%)	6 (5.1%)	3 (4.8%)	0.782
Symptom to ER time (h)	4.7 ± 5.2	4.8 ± 5.1	4.3 ± 5.1	5.0 ± 6.0	0.615
Clinical presentation					0.223
NSTEMI (%)	161 (39.8%)	95 (42.2%)	47 (40.2%)	19 (30.2%)	
STEMI (%)	244 (60.2%)	130 (57.8%)	70 (59.8%)	44 (69.8%)	
Killip class III/IV (%) *	20 (4.9%)	17 (7.6%)	0 (0%)	3 (4.8%)	0.009
SBP (mm Hg)	136.5 ± 28.8	134.5 ± 29.7	141.0 ± 28.1	135.4 ± 26.6	0.132
DBP (mm Hg) *^,$^	80.6 ± 17.2	79.1 ± 17.3	84.9 ± 17.0 *	78.1 ± 15.9 ^$^	0.011
HR (/min)	79.1 ± 20.3	80.0 ± 21.6	79.2 ± 18.5	75.7 ± 18.7	0.376
Chemistry					
TC (mg/dL)	179.1 ± 43.0	177.0 ± 45.1	179.9 ± 40.5	184.9 ± 40.1	0.428
LDL (mg/dL)	117.4 ± 38.0	116.5 ± 40.2	119.5 ± 36.9	116.8 ± 32.0	0.684
HDL (mg/dL)	44.9 ± 11.7	44.9 ± 12.3	44.8 ± 11.3	45.1 ± 10.3	0.888
Cr (mg/dL)	1.07 ± 1.17	1.12 ± 1.12	1.06 ± 1.36	0.90 ± 0.30	0.396
CK-MB (U/L)	2024.9 ± 2135.8	1893.9 ± 2095.4	2194.2 ± 2296.0	2181.1 ± 1966.5	0.132
Troponin-I (ng/L)	46.8 ± 60.6	42.3 ± 59.6	50.0 ± 60.2	57.6 ± 64.0	0.058
Echocardiographic findings					
LVESD (mm)	34.5 ± 7.4	34.8 ± 7.8	34.5 ± 6.7	33.3 ± 7.0	0.416
LVEDD (mm)	47.7 ± 6.6	47.7 ± 7.0	48.2 ± 5.9	46.6 ± 6.9	0.299
LVESV (mL)	51.1 ± 24.6	52.2 ± 27.1	51.8 ± 22.1	49.7 ± 19.3	0.841
LVEDV (mL)	95.6 ± 32.3	95.8 ± 34.3	96.3 ± 31.5	93.3 ± 25.9	0.883
LVEF (%)	47.4 ± 10.6	47.0 ± 11.5	47.9 ± 9.3	47.6 ± 9.4	0.986
WMSI	1.54 ± 0.39	1.55 ± 0.42	1.51 ± 0.35	1.52 ± 0.32	0.850
LA diameter (mm)	37.6 ± 5.7	37.8 ± 6.1	37.8 ± 5.7	36.4 ± 4.1	0.090
LAVI (mL/m^2^)	35.4 ± 16.2	37.0 ± 17.1	34.5 ± 16.6	31.4 ± 9.7	0.084
Mitral E velocity (cm/s)	68.4 ± 22.1	68.5 ± 22.6	70.3 ± 23.9	64.5 ± 16.1	0.253
Mitral A velocity (cm/s)	78.1 ± 21.0	80.1 ± 21.5	76.1 ± 20.8	75.0 ± 19.4	0.084
Mitral E/A ratio	0.92 ± 0.41	0.90 ± 0.40	0.98 ± 0.47	0.90 ± 0.27	0.366
Mitral annular e’ velocity (cm/s)	6.0 ± 2.1	5.9 ± 2.2	6.3 ± 1.9	6.4 ± 1.8	0.084
Mitral annular a’ velocity (cm/s)	8.7 ± 2.2	8.6 ± 2.4	8.6 ± 2.1	9.1 ± 2.0	0.480
E/e’ ratio	12.3 ± 6.2	13.0 ± 7.3	12.0 ± 4.7	10.8 ± 3.9 ^+^	0.103
TR Vmax (m/s)	2.6 ± 0.4	2.6 ± 0.4	2.6 ± 0.4	2.5 ± 0.4	0.327
Culprit vessels	(*n* = 400)				0.095
LMCA	10 (2.5%)	5 (1.3%)	1 (0.9%)	4 (6.3%)	
LAD	198 (49.4%)	102 (46.1%)	68 (58.1%)	28 (44.2%)	
LCX	66 (16.5%)	35 (16.0%)	19 (16.2%)	12 (19.0%)	
RCA	126 (31.6%)	78 (35.6%)	29 (24.8%)	19 (30.2%)	
Pre-TIMI grade	(*n* = 400)				0.046
TIMI 0	211 (52.8%)	119 (54.1%)	54 (46.2%)	38 (60.3%)	
TIMI I	25 (6.3%)	15 (6.8%)	6 (5.1%)	4 (6.3%)	
TIMI II	42 (10.2%)	23 (10.2%)	9 (7.7%)	10 (15.9%)	
TIMI III	121 (30.3%)	62 (28.2%)	48 (41.0%)	11 (17.5%)	
Post-TIMI grade	(*n* = 400)				0.468
TIMI I	1 (0.3%)	1 (0.5%)	0 (0%)	0 (0%)	
TIMI II	14 (3.5%)	9 (4.1%)	5 (4.3%)	0 (0%)	
TIMI III	385 (96.3%)	210 (95.9%)	112 (95.7%)	63 (100.0%)	
Complete revascularization (%)	385 (96.3%)	210 (95.9%)	112 (95.7%)	63 (100.0%)	0.468

* *p*-value < 0.05 between No-CR group and insufficient-CR group, ^+^ *p*-value < 0.05 between No-CR group and CR group, ^$^ *p*-value < 0.05 between insufficient-CR group and CR group. BMI, body mass index; CK-MB, creatine kinase myocardial band; Cr, creatinine; DBP, diastolic blood pressure; ER, emergency room; HDL, high-density lipoprotein; HR, heart rate; HTN, hypertension; LA, left atrium; LAVI, left atrial volume index; LAD, left anterior descending artery; LCX, left circumflex artery; LDL, low-density lipoprotein; LMCA, left main coronary artery; LVEDD, left ventricular end-diastolic dimension; LVEDV, left ventricular end-diastolic volume; LVESD, left ventricular end-systolic dimension; LVESV, left ventricular end-systolic volume; LVEF, left ventricular ejection fraction; MI, myocardial infarction; NSTEMI, non-ST-segment elevation myocardial infarction; RCA, right coronary artery; SBP, systolic blood pressure; STEMI, ST-segment elevation myocardial infarction; TC, total cholesterol; TIMI, thrombolysis in myocardial infarction; TR, tricuspid regurgitation; WMSI, wall motion score index.

**Table 2 jcm-10-02088-t002:** Comparison of baseline and follow-up echocardiographic findings according to the presence of cardiac rehabilitation (CR).

	No-CR Group (*n* = 225)			Insufficient-CR Group (*n* = 117)			CR Group (*n* = 63)			*p* Value *
	Baseline	Follow-Up	*p*-Value	Baseline	Follow-Up	*p*-Value	Baseline	Follow-Up	*p*-Value	
LVESD (mm)	34.8 ± 7.8	35.9 ± 18.1	0.658	34.5 ± 6.7	33.8 ± 7.4	0.208	33.3 ± 7.0	33.3 ± 7.4	0.705	0.445
LVEDD (mm)	47.7 ± 7.0	48.5 ± 7.1	0.042	48.2 ± 5.9	48.8 ± 6.2	0.620	46.6 ± 6.9	47.6 ± 8.1	0.080	0.613
LVESV (mL)	52.2 ± 27.1	49.2 ± 26.4	0.020	51.8 ± 22.1	47.2 ± 24.2	0.006	49.7 ± 19.3	46.0 ± 24.9	0.020	0.621
LVEDV (mL)	95.8 ± 34.3	94.8 ± 32.3	0.837	96.3 ± 31.5	93.2 ± 30.2	0.308	93.3 ± 25.9	95.6 ± 30.0	0.546	0.903
LVEF (%)	47.0 ± 11.5	51.0 ± 12.1	<0.001	47.9 ± 9.3	52.0 ± 10.5	<0.001	47.6 ± 9.4	53.9 ± 11.0	<0.001	0.233
LA diameter (mm)	37.8 ± 6.1	38.2 ± 6.3	0.138	37.8 ± 5.7	38.0 ± 5.2	0.936	36.4 ± 4.1	37.1 ± 4.7	0.310	0.273
LAVI (mL/m^2^)	37.0 ± 17.1	35.1 ± 19.2	0.001	34.5 ± 16.6	31.6 ± 14.2	0.027	31.4 ± 9.7+	29.4 ± 10.6	0.049	0.079
Mitral E velocity (cm/s)	68.5 ± 22.6	63.4 ± 20.1	0.023	70.3 ± 23.9	64.7 ± 26.0	0.006	64.5 ± 16.1	61.6 ± 16.6	0.192	0.574
Mitral A velocity (cm/s)	80.1 ± 21.5	75.9 ± 22.8	0.005	76.1 ± 20.8	73.2 ± 20.5	0.019	75.0 ± 19.4	75.0 ± 17.5	0.795	0.338
E/A ratio	0.90 ± 0.40	0.94 ± 0.63	0.338	0.98 ± 0.47	0.95 ± 0.76	0.024	0.90 ± 0.27	0.88 ± 0.44	0.137	0.812
Mitral annular e’ velocity (cm/s)	5.9 ± 2.2	5.9 ± 2.0	0.906	6.3 ± 1.9	6.1 ± 1.9	0.593	6.4 ± 1.8	6.8 ± 2.3	0.265	0.024
Mitral annular a’ velocity (cm/s)	8.6 ± 2.4	8.6 ± 2.3	0.496	8.6 ± 2.1	9.0 ± 2.3	0.578	9.1 ± 2.0	9.5 ± 2.1	0.106	0.009
E/e’ ratio	13.0 ± 7.3	12.0 ± 7.0	0.612	12.0 ± 4.7	11.1 ± 6.1	0.655	10.8 ± 3.9+	10.4 ± 5.7	0.999	0.009
TR Vmax (m/s)	2.6 ± 0.4	2.7 ± 0.5	0.927	2.6 ± 0.4	2.6 ± 0.5	0.533	2.5 ± 0.4	2.5 ± 0.3	0.109	0.254

* *p*-value comparing baseline and follow-up values among the three groups. LA, left atrium; LAVI, left atrial volume index; LVEDD, left ventricular end-diastolic dimension; LVEDV, left ventricular end-diastolic volume; LVESD, left ventricular end-systolic dimension; LVESV, left ventricular end-systolic volume; LVEF, left ventricular ejection fraction; TR, tricuspid regurgitation.

**Table 3 jcm-10-02088-t003:** Echocardiographic variables of diastolic dysfunction according to the cardiac rehabilitation groups.

	No-CR Group (*n* = 225)	Insufficient-CR Group (*n* = 117)	CR Group (*n* = 63)	*p*-Value
Baseline				
E/e’ ratio > 14	61 (27.1%)	27 (23.1%)	10 (15.9%)	0.174
Septal e’ velocity < 7 cm/s	135 (60.0%)	70 (59.8%)	37 (58.7%)	0.983
LAVI > 34 mL/m^2^	108 (48.0%)	48 (41.0%)	23 (36.5%)	0.191
TR Vmax > 2.8 m/s	38 (16.9%)	20 (17.1%)	11 (17.5%)	0.994
Total number	1.52	1.41	1.29	0.358
Estimation of LV filling pressure				0.408
Normal LV filling pressure	92 (40.9%)	53 (45.3%)	27 (42.9%)	
Grade 1 diastolic dysfunction	77 (34.2%)	38 (32.5%)	27 (42.9%)	
Grade 2 diastolic dysfunction	53 (23.6%)	25 (21.4%)	9 (14.3%)	
Grade 3 diastolic dysfunction	3 (1.3%)	1 (0.9%)	0 (0%)	
Follow-up				
E/E’ ratio > 14	57 (25.3%)	20 (17.1%)	10 (15.9%)	0.106
Septal e’ velocity < 7 cm/s	139 (61.8%)	74 (63.2%)	32 (50.8%)	0.222
LAVI > 34 mL/m^2^	92 (40.9%)	41 (35.0%)	15 (23.8%) ^+^	0.042
TR Vmax > 2.8 m/s	49 (21.8%)	19 (16.2%)	9 (14.3%)	0.270
Total number	1.50	1.32	1.05 ^+^	0.017
Estimation of LV filling pressure				0.083
Normal LV filling pressure	112 (49.8%)	59 (50.4%)	41 (52.3%)	
Grade 1 diastolic dysfunction	54 (24.0%)	37 (31.6%)	13 (20.6%)	
Grade 2 diastolic dysfunction	46 (20.4%)	19 (16.2%)	6 (9.5%)	
Grade 3 diastolic dysfunction	13 (5.8%)	2 (1.7%)	3 (4.8%)	

^+^ *p*-value < 0.05 between No-CR group and CR group.

**Table 4 jcm-10-02088-t004:** Predictors of major adverse clinical event.

Variable	Hazard Ratio	95% Confidential Interval	*p*-Value
Univariate analysis			
Age (year)	1.029	1.015–1.044	<0.001
Male sex	0.988	0.692–1.412	0.949
BMI (kg/m^2^)	0.990	0.938–1.046	0.727
Killip class III/IV	1.390	0.709–2.725	0.338
SBP (mmHg)	1.005	0.999–1.010	0.095
DBP (mmHg)	1.001	0.998–1.007	0.627
HR (/min)	1.008	1.000–1.015	0.052
Cardiac rehabilitation			
No-CR group	Reference		
Insufficient-CR group	0.693	0.477–1.007	0.055
CR group	0.606	0.367–1.000	0.049
STEMI	0.790	0.575–1.084	0.145
Hypertension	1.590	1.158–2.184	0.004
Diabetes	1.675	1.215–2.309	0.002
Smoking	0.819	0.566–1.185	0.290
Creatinine	1.032	0.934–1.141	0.507
CK-MB	1.000	1.000–1.000	0.723
Troponin-I	0.999	0.997–1.002	0.620
Baseline LVESV (mL)	1.000	0.993–1.006	0.884
Baseline LVEF (%)	0.995	0.980–1.010	0.514
Baseline LAVI (mL/m^2^)	1.013	1.004–1.022	0.004
Baseline E/e’ ratio	1.047	1.025–1.069	<0.001
Baseline TR Vmax (m/s)	1.546	1.017–2.349	0.041
No. of baseline diastolic parameters	1.210	1.060–1.382	0.005
Follow-up LVESV (mL)	1.008	1.002–1.014	0.011
Follow-up LVEF (%)	0.988	0.975–1.001	0.066
Follow-up LAVI (mL/m^2^)	1.020	1.012–1.027	<0.001
Follow-up E/e’ ratio	1.032	1.014–1.051	<0.001
Follow-up TR Vmax (m/s)	2.216	1.568–3.132	<0.001
No of follow-up diastolic parameters	1.356	1.198–1.536	<0.001
Multivariate analysis			
Age (year)	1.035	1.018–1.052	<0.001
SBP (mmHg)	1.005	0.999–1.011	0.100
HR (/min)	1.002	0.995–1.010	0.555
Hypertension	1.228	0.869–1.736	0.245
Diabetes	1.627	1.158–2.286	0.005
Follow-up LVESV (mL)	0.999	0.992–1.007	0.883
Follow-up LVEF (%)	0.995	0.984–1.006	0.374
Cardiac rehabilitation			
No-CR group	Reference		
Insufficient-CR group	0.702	0.478–1.031	0.071
CR group	0.738	0.440–1.237	0.249
No of follow-up diastolic parameters	1.255	1.076–1.465	0.004

BMI, body mass index; CK-MB, creatinine kinase MB fraction; DBP, diastolic blood pressure; HR, heart rate; LAVI, left atrial volume index; LVEF, left ventricular ejection fraction; LVESV, left ventricular end-systolic volume; SBP, systolic blood pressure; TR Vmax, maximal velocity of tricuspid regurgitation.

## Data Availability

Data sharing not applicable

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
