# Peer review of "Effect of Cardiac Rehabilitation on Left Ventricular Diastolic Function in Patients with Acute Myocardial Infarction"

_jcm, 2021, doi:10.3390/jcm10102088_

Round 1

Reviewer 1 Report

In this paper, authors investigated the role of cardiac rehabilitation on diastolic parameters of patients with previous myocardial infarction. The topic is interesting and the idea that CR could improve the overall survival of patients affected by AMI is well confirmed in literature. Despite the authors have already well designed the paper, I think that there are major points that could interfere on the results; so I would suggest them to better define the following concerns.

1) the data that the no-CR group will have less improvement of the diastolic parameters in comparison with the CR group could be influenced by two principal differences on baseline, such as age (older in the no-CR group) and arterial hypertension (more patients with arterial hypertension in the no-CR group). It is clear that there was no statistical significance of diastolic parameters among the three groups at the baseline echocardiographic examinations; however, age and arterial hypertension are two predictors of diastolic dysfunction (see for example "Cheng S, et al. Age and the effectiveness of anti-hypertensive therapy on improvement in diastolic function. J Hypertens. 2014 Jan;32(1):174-80" and "Nadruz W, Shah AM, Solomon SD. Diastolic Dysfunction and Hypertension. Med Clin North Am. 2017 Jan;101(1):7-17". I believe that authors have to better reinforce this concept in the Discussion (they said about age that "the active participants to the CR program were younger, they could have lower LAVI and mitral E/e’ ratio", but they did not investigate the arterial hypertension clue. At this purpose, this point could be stressed in relation to the values of blood pressure, as far as it is known that lowering too much the blood pressure could be dangerous in some settings, as demonstrated by "Di Nora C. et al Systolic blood pressure target in systemic arterial hypertension: Is lower ever better? Results from a community-based Caucasian cohort. Eur J Intern Med. 2018 Feb;48:57-63". Thus, I will suggest to add some comments on this in the Discussion.

2) it is not clear which are the parameters in MACE

3) it is not clear when the follow up echocardiograms have been performed after the CR (one week? one month?)

4) in the Background could be useful to cite "Role of Cardiac Rehabilitation After Ventricular Assist Device Implantation. Heart Fail Clin. 2021 Apr;17(2):273-278" to reinforce the role of CR in all the settings of cardiac surgeries

5) any comments about the LVEF with a mean of 47% in all the groups? Any kynetic abnormalities?

Author Response

In this paper, authors investigated the role of cardiac rehabilitation on diastolic parameters of patients with previous myocardial infarction. The topic is interesting and the idea that CR could improve the overall survival of patients affected by AMI is well confirmed in literature. Despite the authors have already well designed the paper, I think that there are major points that could interfere on the results; so I would suggest them to better define the following concerns.

1) the data that the no-CR group will have less improvement of the diastolic parameters in comparison with the CR group could be influenced by two principal differences on baseline, such as age (older in the no-CR group) and arterial hypertension (more patients with arterial hypertension in the no-CR group). It is clear that there was no statistical significance of diastolic parameters among the three groups at the baseline echocardiographic examinations; however, age and arterial hypertension are two predictors of diastolic dysfunction (see for example "Cheng S, et al. Age and the effectiveness of anti-hypertensive therapy on improvement in diastolic function. J Hypertens. 2014 Jan;32(1):174-80" and "Nadruz W, Shah AM, Solomon SD. Diastolic Dysfunction and Hypertension. Med Clin North Am. 2017 Jan;101(1):7-17". I believe that authors have to better reinforce this concept in the Discussion (they said about age that "the active participants to the CR program were younger, they could have lower LAVI and mitral E/e’ ratio", but they did not investigate the arterial hypertension clue. At this purpose, this point could be stressed in relation to the values of blood pressure, as far as it is known that lowering too much the blood pressure could be dangerous in some settings, as demonstrated by "Di Nora C. et al Systolic blood pressure target in systemic arterial hypertension: Is lower ever better? Results from a community-based Caucasian cohort. Eur J Intern Med. 2018 Feb;48:57-63". Thus, I will suggest to add some comments on this in the Discussion.

Thank you for your valuable comment. We agree with your opinion. Thus, we revised our manuscript as your recommendation. Because we did not check the control status of blood pressure after anti-ischemic therapy and CR, we did not know the effect of the follow-up blood pressure on the long-term clinical outcome. So, we did not inserted the reference that you recommended (Di Nora C. et al Systolic blood pressure target in systemic arterial hypertension: Is lower ever better? Results from a community-based Caucasian cohort. Eur J Intern Med. 2018 Feb;48:57-63) into the manuscript. Thank you very much for your comment.

Before revision in the Discussion section

Our study showed baseline LAVI and mitral E/e’ ratio were the lowest in the CR group. These differences may come from the age difference of the study groups. Because the active participants to the CR program were younger, they could have lower LAVI and mitral E/e’ ratio. Although there were significant differences of these diastolic parameters, the number of echocardiographic diastolic parameters was similar among the study groups.

After revision in the Discussion section

Our study showed baseline LAVI and mitral E/e’ ratio were the lowest in the CR group. These differences may come from the age difference of the study groups. Also, age is another determinant of the worsening of diastolic function, and increased prevalence of LV diastolic dysfunction can be associated with increasing age [19]. Because the active participants to the CR program were younger, they could have lower prevalence of hypertension, lower LAVI and mitral E/e’ ratio. Although there were significant differences of these diastolic parameters, the number of echocardiographic diastolic parameters was similar among the study groups. Younger age of the CR group can affect the change of LAVI. Cheng S et al. showed younger patients had better improvement of diastolic parameters in response to similar reduction in systolic blood pressure [20].

2) it is not clear which are the parameters in MACE

Thank you for your comment. We included deaths, recurrence of MI, angina and revascularization, admissions for heart failure, and stroke or transient ischemic attack as MACE. Thus, we revised our manuscript as your recommendation.

Before revision in the Methods section

We checked for incidence of death in the medical records of patients who had been regularly followed-up. In patients without regular follow-ups, we identified major adverse cardiac events (MACE) by speaking with the patients or their relatives over telephone, or data from the Korean national insurance service. 

After revision in the Methods section

We checked for incidence of death in the medical records of patients who had been regularly followed-up. In patients without regular follow-ups, we identified major adverse cardiac events (MACE) including deaths, recurrence of MI, angina and revascularization, admissions for heart failure, and stroke or transient ischemic attack by speaking with the patients or their relatives over telephone, or data from the Korean national insurance service.

3) it is not clear when the follow up echocardiograms have been performed after the CR (one week? one month?)

Thank you for your comment. We checked the interval between the last CR and the follow-up echocardiography and the intervals between the last CR and the follow-up echocardiography was 17.1 ± 14.9 months in the insufficient CR group and 13.4 ± 11.5 months in the CR group. So, we inserted these into the results section.

Before revision in the Methods section

Follow-up echocardiographic examinations were performed for a mean duration of 18.0 ± 15.1 months.

After revision in the Methods section

Follow-up echocardiographic examinations were performed for a mean duration of 18.0 ± 15.1 months, and the intervals between the last CR and the follow-up echocardiography was 17.1 ± 14.9 months in the insufficient-CR group and 13.4 ± 11.5 months in the CR group.

4) in the Background could be useful to cite "Role of Cardiac Rehabilitation After Ventricular Assist Device Implantation. Heart Fail Clin. 2021 Apr;17(2):273-278" to reinforce the role of CR in all the settings of cardiac surgeries.

Thank you very much for your comment. We think the review article by Di Nora C et al. is the best review article about the role of CR in the patients with HF especially treated with VAD. We inserted the article into the 10th reference.

Before revision in the Methods section

The CR-associated exercise program can improve cardiac function in patients with coronary artery diseases [7-9].

After revision in the Methods section

The CR-associated exercise program can improve cardiac function in patients with coronary artery diseases [7-9]. CR can improve clinical outcomes also in patients with heart failure (HF), including end-stage HF treated with a left ventricular assist device system [10]

5) any comments about the LVEF with a mean of 47% in all the groups? Any kynetic abnormalities?

Thank you for your comment. Do you mean ‘kinetic abnormality’ as regional wall motion abnormality? There was no statistical difference of regional wall motion abnormality among groups. We mentioned about LVEF and wall motion score index in the results section. Also, there was no statistical difference among groups at the follow-up echocardiography (1.40 ± 0.43 vs. 1.38 ± 0.37 vs. 1.35 ± 0.33, P=0.722).

Before revision in the Methods section

Regarding echocardiographic parameters, there were no statistically significant differences of diastolic parameters among three groups except LAVI (P=0.043) and E/e’ ratio (P=0.045).

After revision in the Methods section

Regarding echocardiographic parameters, there was no statistical difference of LVEF (47.0 ± 11.5% vs. 47.9 ±9.3% vs. 47.6 ± 9.4%, P=0.986) and regional wall motion abnormality assessed by wall motion score index (1.55 ± 0.42 vs. 1.51 ± 0.35 vs. 1.52 ± 0.32, P=0.850). Also, there were no statistically significant differences of diastolic parameters among three groups except LAVI (P=0.043) and E/e’ ratio (P=0.045).

Reviewer 2 Report

The authors analyzed change of diastolic function and its prognostic impact after cardiac rehabilitation in patients with acute myocardial infarction. Although the aim of the study might have interest, the findings presented in the study remains controversial. Statistically significant differences in the findings can be attributed to the statistically significant difference in the average ages of the groups. Moreover, statistical analyses remain unreliable in the study. The authors do not test the distribution of the samples but perform ANOVA which is a parametric test and used for the normally distributed data. Shapiro-Wilk test should be performed before deciding the appropriate test and then a parametric and non-parametric test can be chosen to analyze the data. A post-hoc test should follow the analyze differences between the groups separately again depending on the distribution of the data. Also t-test may not be sufficient to analyze the follow-up echocardiographic parameters as the distribution of the data should be evaluated first and then t-test or Mann-Whitney U test can be performed.

The authors can assess the normality of the data and perform appropriate statistical tests before evaluating the manuscript again.

Author Response

Second reviewer

The authors analyzed change of diastolic function and its prognostic impact after cardiac rehabilitation in patients with acute myocardial infarction. Although the aim of the study might have interest, the findings presented in the study remains controversial. Statistically significant differences in the findings can be attributed to the statistically significant difference in the average ages of the groups. Moreover, statistical analyses remain unreliable in the study. The authors do not test the distribution of the samples but perform ANOVA which is a parametric test and used for the normally distributed data. Shapiro-Wilk test should be performed before deciding the appropriate test and then a parametric and non-parametric test can be chosen to analyze the data. A post-hoc test should follow the analyze differences between the groups separately again depending on the distribution of the data. Also t-test may not be sufficient to analyze the follow-up echocardiographic parameters as the distribution of the data should be evaluated first and then t-test or Mann-Whitney U test can be performed.

The authors can assess the normality of the data and perform appropriate statistical tests before evaluating the manuscript again.

Thank you very much for your excellent comment. Because we had more than 30 patients in each group, we did an ANOVA test among three groups. But, we checked the distribution and re-analyzed our data as your recommendation. We found several changes and revised our manuscript. We are very appreciating your valuable comment.

Before revision in the Methods section

We compared categorical variables using chi-square test and continuous ones using an analysis of variance (ANOVA) test among the three groups. Moreover, the baseline and the follow-up echocardiographic parameters were compared using a paired-sample t-test.

After revision in the Methods section

We compared categorical variables using chi-square test of categorical variables. For continuous variables, we checked the distribution of the variables by the Shapiro-Wilk test. If the variable showed normal distribution, we used an analysis of variance (ANOVA) test to find statistical differences among the three groups and a t-test between the two groups. We used the Kruskal-Wallis test among three groups and the Mann-Whitney test between the two groups of the variables without normal distribution. Moreover, the baseline and the follow-up echocardiographic parameters were compared using the Wilcoxon singed ranks test.

We revised tables according to your comments.

Table 1. Comparison of clinical and echocardiographic parameters according to the presence of cardiac rehabilitation (CR) therapy in discharged patients

Variable

Total

(n=405)

No-CR group

(n=225)

Insufficient-CR group

(n=117)

CR group

(n=63)

P value

Age (year)+

63.7 ± 11.7

65.2 ± 12.4+

62.9 ± 11.2

61.4± 9.5

0.006

Male sex (%)

300 (74.1%)

164 (72.9%)

90 (76.9%)

46 (73.0%)

0.706

BMI (kg/m2)

24.0 ± 3.0

23.9 ± 3.3

24.1 ± 2.7

23.9 ± 3.0

0.747

Cardiovascular risk factors

HTN (%)+

191 (47.2%)

118 (52.4%)

52 (44.4%)

21 (33.3%)

0.021

DM (%)

127 (31.4%)

79 (35.1%)

33 (28.2%)

15 (23.8%)

0.159

Dyslipidemia (%)

18 (4.4%)

9 (4.0%)

7 (6.0%)

2 (3.2%)

0.722

Smoking (%)

159 (39.5%)

86 (38.6%)

49 (41.9%)

24 (38.1%)

0.415

Prior MI (%)

29 (7.2%)

20 (8.9%)

1 (1.3%)

8 (7.9%)

0.251

Ischemic heart disease (%)

37 (9.1%)

23 (10.2%)

10 (8.5%)

4 (6.3%)

0.619

Family history (%)

17 (4.1%)

8 (3.5%)

6 (5.1%)

3 (4.8%)

0.782

Symptom to ER time (hour)

4.7 ± 5.2

4.8 ± 5.1

4.3 ± 5.1

5.0 ± 6.0

0.615

Clinical presentation

0.223

NSTEMI (%)

161 (39.8%)

95 (42.2%)

47 (40.2%)

19 (30.2%)

STEMI (%)

244 (60.2%)

130 (57.8%)

70 (59.8%)

44 (69.8%)

Killip class III/IV (%)*

20 (4.9%)

17 (7.6%)

0 (0%)

3 (4.8%)

0.009

SBP (mm Hg)

136.5 ± 28.8

134.5 ± 29.7

141.0 ± 28.1

135.4 ± 26.6

0.132

DBP (mm Hg)*,$

80.6 ± 17.2

79.1 ± 17.3

84.9 ± 17.0

78.1 ± 15.9

0.011

HR (/min)

79.1 ± 20.3

80.0 ± 21.6

79.2 ± 18.5

75.7 ± 18.7

0.376

Chemistry

TC (mg/dL)

179.1 ± 43.0

177.0 ± 45.1

179.9 ± 40.5

184.9 ± 40.1

0.428

LDL (mg/dL)

117.4 ± 38.0

116.5 ± 40.2

119.5 ± 36.9

116.8 ± 32.0

0.684

HDL (mg/dL)

44.9 ± 11.7

44.9 ± 12.3

44.8 ± 11.3

45.1 ± 10.3

0.888

Cr (mg/dL)

1.07 ± 1.17

1.12 ± 1.12

1.06 ± 1.36

0.90 ± 0.30

0.396

CK-MB (U/L)

2024.9 ± 2135.8

1893.9 ± 2095.4

2194.2 ± 2296.0

2181.1 ± 1966.5

0.132

Troponin-I (ng/L)*

46.8 ± 60.6

42.3 ± 59.6

50.0 ± 60.2

57.6 ± 64.0

0.058

Echocardiographic findings

 LVESD (mm)

34.5 ± 7.4

34.8 ± 7.8

34.5 ± 6.7

33.3 ± 7.0

0.416

 LVEDD (mm)

47.7 ± 6.6

47.7 ± 7.0

48.2 ± 5.9

46.6 ± 6.9

0.299

 LVESV (ml)

51.1 ± 24.6

52.2 ± 27.1

51.8 ± 22.1

49.7 ± 19.3

0.841

 LVEDV (mL)

95.6 ± 32.3

95.8 ± 34.3

96.3 ± 31.5

93.3 ± 25.9

0.883

LVEF (%)

47.4 ± 10.6

47.0 ± 11.5

47.9 ± 9.3

47.6 ± 9.4

0.986

WMSI

1.54 ± 0.39

1.55 ± 0.42

1.51 ± 0.35

1.52 ± 0.32

0.850

LA diameter (mm)

37.6 ± 5.7

37.8 ± 6.1

37.8 ± 5.7

36.4 ± 4.1

0.090

LAVI (mL/m2)

35.4 ± 16.2

37.0 ± 17.1

34.5 ± 16.6

31.4 ± 9.7+

0.084

Mitral E velocity (cm/s)

68.4 ± 22.1

68.5 ± 22.6

70.3 ± 23.9

64.5 ± 16.1

0.253

Mitral A velocity (cm/s)

78.1 ± 21.0

80.1 ± 21.5

76.1 ± 20.8

75.0 ± 19.4

0.084

 Mitral E/A ratio

0.92 ± 0.41

0.90 ± 0.40

0.98 ± 0.47

0.90 ± 0.27

0.366

Mitral annular e’ velocity (cm/s)

6.0 ± 2.1

5.9 ± 2.2

6.3 ± 1.9

6.4 ± 1.8

0.084

Mitral annular a’ velocity (cm/s)

8.7 ± 2.2

8.6 ± 2.4

8.6 ± 2.1

9.1 ± 2.0

0.480

E/e’ ratio

12.3 ± 6.2

13.0 ± 7.3

12.0 ± 4.7

10.8 ± 3.9+

0.103

TR Vmax (m/s)

2.6 ± 0.4

2.6 ± 0.4

2.6 ± 0.4

2.5 ± 0.4

0.327

Culprit vessels

(n=400)

0.095

 LMCA

10 (2.5%)

5 (1.3%)

1 (0.9%)

4 (6.3%)

 LAD

198 (49.4%)

102 (46.1%)

68 (58.1%)

28 (44.2%)

 LCX

66 (16.5%)

35 (16.0%)

19 (16.2%)

12 (19.0%)

 RCA

126 (31.6%)

78 (35.6%)

29 (24.8%)

19 (30.2%)

Pre-TIMI grade

(n=400)

0.046

 TIMI 0

211 (52.8%)

119 (54.1%)

54 (46.2%)

38 (60.3%)

 TIMI I

25 (6.3%)

15 (6.8%)

6 (5.1%)

4 (6.3%)

 TIMI II

42 (10.2%)

23 (10.2%)

9 (7.7%)

10 (15.9%)

 TIMI III

121 (30.3%)

62 (28.2%)

48 (41.0%)

11 (17.5%)

Post-TIMI grade

(n=400)

0.468

 TIMI I

1 (0.3%)

1 (0.5%)

0 (0%)

0 (0%)

 TIMI II

14 (3.5%)

9 (4.1%)

5 (4.3%)

0 (0%)

 TIMI III

385 (96.3%)

210 (95.9%)

112 (95.7%)

63 (100.0%)

Complete revascularization (%)

385 (96.3%)

210 (95.9%)

112 (95.7%)

63 (100.0%)

0.468

*P value <0.05 between No-CR group and insufficient-CR group, +P value <0.05 between No-CR group and CR group, $P value <0.05 between insufficient-CR group and CR group

BMI, body mass index; CK-MB, creatine kinase myocardial band; Cr, creatinine; DBP, diastolic blood pressure; ER, emergency room; HDL, high-density lipoprotein; HR, heart rate; HTN, hypertension; LA, left atrium; LAVI: left atrial volume index; LAD, left anterior descending artery; LCX, left circumflex artery; LDL, low-density lipoprotein; LMCA, left main coronary artery; LVEDD, left ventricular end-diastolic dimension; LVEDD, left ventricular end-diastolic volume; LVESD, left ventricular end-systolic dimension; LVESV, left ventricular end-systolic volume; LVEF, left ventricular ejection fraction; MI, myocardial infarction; NSTEMI, non-ST-segment elevation myocardial infarction; RCA, right coronary artery; SBP, systolic blood pressure; STEMI, ST-segment elevation myocardial infarction; TC, total cholesterol; TIMI, Thrombolysis in Myocardial Infarction; TR: tricuspid regurgitation

After revision in the Results section

RESULTS

Characteristics of study population

After reviewing all consecutive type 1 MI patients from January 2012 to October 2015, we included 405 patients with baseline and follow-up echocardiographic examinations. The mean interval between the admission date to echocardiographic examinations was 1.6 ± 1.8days (interval: 0 – 14 days).

We divided our study subjects into three groups depending on whether they received CR; No-CR group (n=225), insufficient-CR group (n=117), and CR group (n=63). Their baseline clinical and echocardiographic parameters are expressed in Table 1. Age was significantly higher in the No-CR group (65.2 ± 12.4 years vs. 62.9 ± 11.2 years vs. 61.4 ± 9.5, P=0.006). For cardiovascular risk factors, only hypertension (52.4% vs. 44.4% vs. 33.3%, P=0.021) was significantly higher in the No-CR group. The percentage of non-ST-segment elevation myocardial infarction was similar in the two groups (42.2% vs. 40.2% vs. 30.2%, P=0.223). However, patients with Killip class III/IV were more frequent in the No-CR group (7.6% vs. 0% vs. 4.8%, P=0.009).

Regarding echocardiographic parameters, there was no statistical difference of LVEF (47.0 ± 11.5% vs. 47.9 ±9.3% vs. 47.6 ± 9.4%, P=0.986) and regional wall motion abnormality assessed by wall motion score index (1.55 ± 0.42 vs. 1.51 ± 0.35 vs. 1.52 ± 0.32, P=0.850). Also, there were no statistically significant differences of diastolic parameters among three groups. Coronary angiography and percutaneous coronary intervention were performed in 400 patients (98.6%). Left anterior descending (LAD) coronary artery was the most common culprit lesion. Complete revascularization was achieved in 385 patients (96.3%) and there was no statistical difference in the success rate among the three groups.

Follow-up echocardiography

Follow-up echocardiographic examinations were performed for a mean duration of 18.0 ± 15.1 months, and the intervals between the last CR and the follow-up echocardiography was 17.1 ± 14.9 months in the insufficient CR group and 13.4 ± 11.5 months in the CR group. Findings of the follow-up echocardiography and a comparison between the baseline and follow-up echocardiography findings are summarized in Table 2. At follow-up echocardiography, LVEF was significantly improved in all three groups (No-CR group: 47.0 ± 11.5% to 51.0 ± 12.1%, P<0.001, insufficient-CR group: 47.9 ± 9.3% to 52.0 ± 10.5%, P<0.001, CR group: 47.6 ± 9.4% to 53.9 ± 11.0%, P<0.001). LV end-diastolic dimension was significantly increased in the No-CR group (47.7 ± 7.0 to 48.5 ± 7.1%, P=0.042). LAVI was significantly decreased in all the three groups (No-CR group: 37.0 ± 17.1mL/m2 to 35.1 ± 19.2mL/m2, P=0.001, insufficient-CR group: 34.5 ± 16.6mL/m2 to 31.6 ± 14.2mL/m2, P=0.027, and CR group: 31.4 ± 9.7mL/m2 to 29.4 ± 10.6mL/m2, P=0.049). In addition, mitral E velocity was decreased in the No-CR group (68.5 ± 22.6cm/s to 63.4 ± 20.1cm/s, P=0.023) and the insufficient-CR group (70.3 ± 23.9cm/s to 64.7 ± 26.0cm/s, P=0.006). Mitral A velocity was decreased in the No-CR group (80.1 ± 21.5cm/s to 75.9 ± 22.8cm/s, P=0.005) and the insufficient-CR group (76.1 ± 20.8cm/s to 73.2 ± 20.5cm/s, P=0.019). However, mitral E and A velocity did not change in the CR group (P=0.192 and P=0.795, respectively).

In the comparison of the three groups, mitral annular e’ and a’ velocities were higher in the CR group (P=0.024, and P=0.009, respectively) and mitral E/e’ ratio was significantly lower (P=0.009) in the CR group.

Echocardiographic variables of diastolic dysfunction

Table 3 describes the presence of echocardiographic variables of diastolic dysfunction in the three groups. There was no statistical significance of diastolic parameters among the three groups at the baseline echocardiographic examinations including estimation of LV filling pressures. At the follow-up echocardiographic examinations, patients with LAVI >34mL/m2 was significantly higher in the CR group (P=0.042). Also, total number of diastolic variables was significantly lower in the CR group (P=0.017). The statistical differences of LAVI > 34mL/m2 and total number of diastolic parameters mainly occurred between the No-CR and CR groups (P=0.018 and P=0.006, respectively). At the follow-up echocardiographic examinations, the presence of normal LV filling pressure was higher in the CR group. However, there was a marginal statistical significance (P=0.083).

DISCUSSION

In this study, we compared the follow-up and the baseline echocardiographic variables in patients with AMI. We showed that the CR group had statistically the highest mitral e’ and a’ velocities, the lowest mitral E/e’ ratio, and the lowest total number of echocardiographic parameters of diastolic dysfunction at the follow-up echocardiographic examinations in patients with AMI.

Thank you very much for your comments. Because of your comments, our manuscript became a better one.

Round 2

Reviewer 1 Report

The paper has been improved after the revisions.

Reviewer 2 Report

The authors responded to the reviewer's comments sufficiently. The submitted manuscript can be accepted as a publication.

This manuscript is a resubmission of an earlier submission. The following is a list of the peer review reports and author responses from that submission.

Round 1

Reviewer 1 Report

This is a study about the evaluation of echocardiographic parameters of diastolic dysfunction in patients with previous MI undergoing or not cardiac rehabilitation. The study has several limitations starting from the selection criteria used to define MI. Secondly echocardiographic parameters of diastolic dysfunction have been mainly considered singularly, where the real meaning of these variables derives from the overall view of them ( as suggested by several recognized algorithms). Therefore I have several comments:

  • The definition used for myocardial infarction is quite inaccurate. Was the fourth universal definition of MI used (PMID: 30153967)? Have specific ICD codes been searched for MI? The criteria used by the authors are vague and considering the retrospective nature of the study cases of heart failure might have been selected.
  • Is is not precise to say that LV systolic function was derived by EF. EF is only a part of the systolic function. I would better write: Ejection fraction was calculate by biplane Simpson’s method.
  • I would remove this sentence “This section may be divided by subheadings. It should provide a concise and precise description of the experimental results, their interpretation, as well as the experimental conclusions that can be drawn.”
  • When was the basal echocardiogram performed? Before discharge from the acute phase of MI? the first days after the MI? Diastolic function significantly change in days around an acute phase of an MI.
  • It is not possible to see the whole table 1 in the pdf format.
  • To define the diastolic disfunction degree and analyze it would have been better to stratify the population according to presence or not of increased filling pressure (PMID: 27422899) rather than to consider each single variable.
  • How can author use average and standard deviation for parameter as E’? was a normal distribution present? Was not better to use a median and non-parametric test to compare among groups?
  • Was echocardiography available for all the patients included? It seems to be really hard considering the retrospective nature of the study. Please specify the number of echo analyzed.
  • Was the localization of MI considered in the multivariate analysis? A reduced lateral E’ might be related to a lateral MI.
  • How was mitral anulus E’ calculated? Medial e’? lateral e’? average of the two values?
  • Why LAVI should decrease in the noCR group?
  • What does it mean “Also, total number of diastolic variables was significantly lower in the CR group (P=0.049). The differences of LAVI > 34mL/m2 and total number were seen in the No-CR and CR groups (P=0.013, P=0.048, respectively)? This meaning of this sentence is arguable. Please rephrase.
  • Please state in methods which are the endpoints of the study. Why to evaluate MACE?
  • Why to analyze the total number of diastolic parameters at follow-up? Did you analyze those with parameters indicative of diastolic dysfunction? As it is written it seems that if a measure more parameter of diastolic disfunction ( independently by the results) I have more MACE.
  • Please quote PMID: 33271207 stating also the role of physical performance and frailty in the assessment of diastolic dysfunction.
  • The discussion did not justify the results. Why should I expect a reduction in LAVI of patients with no CR? And why should it increase in a person with CR ( this is not considered such competitive sporting activity).

Author Response

Thank you for your detailed comment. I attached the file including the list of revision. 

Reviewer 2 Report

The authors present a well-written manuscript concerning diastolic echocardiographic parameters in patients undergoing various degrees of cardiac rehabilitation (CR). The effects of CR are certainly worthy of study.

However, I have the following concerns about this study and analysis:

- I disagree strongly with the statement in the introduction ‘Cardiac rehabilitation (CR) is the best modality associated with improvement of symptoms and survival in patients with AMI’. Surely revascularisation and pharmacotherapy has a stronger effect on symptoms and survival than CR?

- A fundamental problem with this study is the issue of what factors led to the patients completing, incompletely attending or not attend cardiac rehabilitation. Presumably this was down to patient choice, in which case there are likely confounders, even after adjustment for ‘hard’ factors. Those more motivated to complete CR will likely also be those more likely to be compliant with other treatments, lifestyle advice, engagement with medical services etc. I do accept that performing a randomised study is likely unethical given the proven benefits of CR, but this limitation should be strongly emphasised in the manuscript.

- Similarly, the different sample sizes and baseline characteristics in the 3 groups make it difficult to interpret the significance of baseline to follow up changes and comparisons between groups respectively. For example,  A better approach may have been to have derived a cohort of matched controls to compare to the CR group.

- I would question whether the inclusion of the ‘incomplete CR’ group adds anything to the paper as this is presumably a heterogeneous group, some of whom had hardly any CR and some who had almost the full amount.

- It is unclear what the pre-specified primary hypothesis being tested was. There are a lot of variables included and a lot of statistical tests so I worry that the findings around LV diastolic function may be related to type 1 error.

- Re ‘The percentage of non-ST-segment elevation myocardial infarction was similar in the two groups (42.7% vs. 122 41.7% vs. 31.3%, P=0.248)’. Though not statistically significant, I think it is not  describe a 10% difference as ‘similar’.

- The first paragraph of the Results section has been included in error.

- In the results, the focus of the study in on diastolic function but these are addressed in the text after quite a lot of discussion of those relating to systolic function. Perhaps these should be reversed?

Author Response

(The authors gave the same response as above.)

Round 2

Reviewer 1 Report

The author replied to all my questions. I still have major concerns:

  • The analysis according diastolic disfunction degree is not powered enough (there are altogether 8 groups). The analysis of diastolic disfunction considering filling pressure is not statistically significant, thus it is not clear which is the message of the study. As I have already underlined in the last revision the analysis of diastolic disfunction is not made on a single number but on the overall view of variable  E/A together with E’, left atrial volume, systolic pulmonary pressure.
  • Secondly the clinical meaning of the number of diastolic variables available in a retrospective study is arguable.

Author Response

The analysis of diastolic disfunction considering filling pressure is not statistically significant, thus it is not clear which is the message of the study.

Thank you for your comment. We totally understand the review’s concerns. This is a clear limitation of our study. However, it was also originated from the limitations of real clinical practice. Among a total 405 patients, only 63 patients (16%) were classified into sufficient-CR group. As such, the implementation of CR is definitely hard goal in the real world. Therefore, it was difficult to collect a statistically sufficient sample size because 63 patients with sufficient-CR were sub-classified according to the LV diastolic dysfunction grades (0 to 3). We think that the potential readers of this article who are mostly clinicians would consider this kind of limitation. In addition, if CR is more widely implemented in clinical practice in the future, we can obtain more robust data.

As I have already underlined in the last revision the analysis of diastolic disfunction is not made on a single number but on the overall view of variable E/A together with E’, left atrial volume, systolic pulmonary pressure. Secondly the clinical meaning of the number of diastolic variables available in a retrospective study is arguable.Thank you for your comment. We basically classified the diastolic dysfunction grades according to the current ASE guideline, and obtained ‘statistically insufficient’ results as mentioned above. Therefore, we additionally presented another approach to the LV diastolic dysfunction with additive manner of each criterion. Each of the criteria we used (LAVI > 34 mL, TR velocity > 2.8 m/s, and E/e' ratio > 14) was derived from the ASE guideline.  Also, the guideline’s algorithm suggests if more criterion is met, it is closer to more advanced diastolic dysfunction. Considering the limitations of clinical materials, it would be also valuable to consider an additional approach to LV diastolic function. We inserted this into the limitation section. (We performed many statistical analysis to show the difference. However, we did not find the statistical difference. Thus, we used this approach. We are very sorry for that.)

Before revision in the limitation section

Second, we did not exclude other factors, including significant valvular heart disease, which can affect LAVI.

After revision in the Limitation section

Second, we used the sum of diastolic parameters in our study. This approach has not been used in previous studies. Because the guidelines’ algorithm suggests that more criteria are met, it is closer to have more advanced diastolic dysfunction. It should be validated in other studies. Third, we did not exclude other factors, including significant valvular heart disease, affecting LAVI.

Thank you very much for your review.

Reviewer 2 Report

Thanks for revising this manuscript. My comments have been addressed.

Author Response

Thank you very much for your kind review.